# Poliomyelitis in Nigeria: Impact of Vaccination Services and Polio Intervention and Eradication Efforts

**DOI:** 10.3390/vaccines13030232

**Published:** 2025-02-25

**Authors:** Obinna V. Eze, Johanna C. Meyer, Stephen M. Campbell

**Affiliations:** 1School of Health Sciences, University of Manchester, Manchester M13 9PL, UK; obinna.eze@mft.nhs.uk; 2Manchester University NHS Foundation Trust, Oxford Road, Manchester M13 9WL, UK; 3Department of Public Health Pharmacy and Management, School of Pharmacy, Sefako Makgatho Health Sciences University, Ga-Rankuwa, Pretoria 0208, South Africa; hannelie.meyer@smu.ac.za; 4South African Vaccination and Immunisation Centre, Sefako Makgatho Health Sciences University, Ga-Rankuwa, Pretoria 0208, South Africa

**Keywords:** poliomyelitis, Nigeria, vaccination programs, eradication, vaccine hesitancy, wild poliovirus (WPV), circulating vaccine-derived poliovirus (cVDPV), paralytic polio, immunization, global burden

## Abstract

**Background**: Polio is an infectious viral disease that can cause paralytic complications and death. Despite global efforts to eradicate wild poliovirus, there are ongoing outbreaks globally and the mutated form of paralytic polio, i.e., circulating vaccine-derived poliovirus, is present in Nigeria. Low vaccination uptake and poor sanitation are responsible for outbreaks in countries where polio had previously been eliminated. This review identifies policies, strategies and interventions for polio eradication and assesses their impact on polio vaccine uptake and eradication efforts in Nigeria. **Methods**: A systematic literature review was conducted and guided by the Population, Intervention, Comparator and Outcome (PICO) framework and Preferred Reporting Items for Systematic Reviews and Meta-Analyses (PRISMA) flowchart, with identified articles appraised using the Critical Appraisal Skills Program appraisal tool. **Results**: A total of 393 articles were identified, of which 26 articles were included. Key findings indicate polio intervention services, policies and mass campaigns have had a significant impact on eradicating WPV in Nigeria. However, there are gaps in variant polio eradication efforts, with low vaccination uptake, poor surveillance, vaccine hesitancy, lack of community engagement, weaknesses in the healthcare system and other challenges in Nigeria regionally and nationally, posing a risk to public health that threatens the eradication of all forms of polio in Nigeria. **Conclusions**: Recommendations are suggested for changes to practice and policy to improve polio vaccination uptake in Nigeria and globally in the short-term (1–2 years), mid-term (3–4 years) and long-term (5+ years). Collaborative targeted polio vaccination programs and funding of public health infrastructure are imperative globally alongside national strategic policy intervention frameworks to strengthen the World Health Organization Global Polio Eradication Initiative and improve vaccine uptake and monitoring of vaccine hesitancy. Simultaneous health-literate community engagement is needed to achieve and maintain polio eradication efforts, which must be integrated into national health frameworks and coordinated across the African continent.

## 1. Introduction

Polio has been a public health threat since the 20th century. It is a highly infectious viral disease with severe complications and is prone to outbreaks [1,2]. Polio primarily affects children, with symptoms including low fever, sore throat, severe limb pain, flaccid paralysis, limb deformities and permanent disability [2,3]. A sore throat and low fever are the hallmarks of the clinical symptoms in most cases, which often go away in less than a week [4]. Paralytic polio occurs in <1% of infections; it can be permanent and present 12 months after onset [5], with 5–10% of paralytic polio cases resulting in death [6,7,8]. Paralytic polio cases reached epidemic levels in the 20th century, after the first outbreaks in Europe and the United States were reported in the 19th century [1,9]. Global estimates show that 12–20 million people have been impacted by the infection since the 19th century [3,7].

In 1988, the Global Polio Eradication Initiative (GPEI) was launched after a resolution passed by the World Health Assembly, with over 350,000 children recorded as having paralytic polio across 125 countries [10]. The GPEI’s goals are to detect and stop the spread of the poliovirus and strengthen immunization programs globally. Post-eradication strategies, such as sustaining high vaccination rates and strengthening surveillance efforts, are important in the goal of eradicating polio [11,12]. In 2014, as part of the GPEI, the World Health Organization (WHO) recommended adding at least one dose of inactivated polio vaccine (IPV) to routine immunization schedules. This measure was proposed as a strategy to reduce the potential risk of re-emergence of type 2 polio after the Sabin type 2 strains have been removed from the oral polio vaccine (OPV) [9,10,13,14]. The GPEI Polio Eradication Strategy 2022–2026 set 2023 as a target year to interrupt all remaining type 1 wild poliovirus (WPV1) transmission (Goal One) and type 2 circulating vaccine-derived poliovirus (cVDPV) transmission (Goal Two), with the aim of reaching eradication by 2026.

From January 2021 to May 2023, only one of the three WPV serotypes, i.e., WPV1, was circulating in four countries: Afghanistan, Pakistan, Namibia and Mozambique [15]. Currently, WPV1 remains endemic in Afghanistan and Pakistan [6,16], with a mutated variant form of paralytic polio, i.e., cVDPV, in 32 countries, including Nigeria [17,18,19,20]. In August 2024, the United Nations reported a case of paralytic polio in Gaza for the first time in the 25 years of its polio-free status, with conflict disrupting regular child vaccination programs [21,22]. The infection was linked to the type 2 variant detected earlier in environmental samples collected from Gaza wastewater, with poor sanitation resulting in poliovirus typically spreading through the fecal–oral route in unsanitary environments [15,23,24].

While Nigeria, the seventh most populous country in the world, was declared free of WPV in 2020, the transmission of the circulating vaccine-derived poliovirus type 2 (cVPV2) strain and eradication of all variant strains of polio remain a concern [25]. As of August 2024, 30 cases of cVDPV have been reported in Nigeria, constituting 42% of the 72 cases of cVDPV reported globally, with the potential to amplify the transmission through a cross-border spread [26,27].

Nigeria has focused on the management of infectious diseases and outbreaks such as Ebola, Lassa fever and COVID-19 using the incident management approach with support provided by the Nigerian Ministry of Health, Nigerian Center for Disease Control, private sector and international community [28].

Several interventions and policies have been established for the eradication of all forms of polio in Nigeria; for example, the 1999 National Program on Immunization (NPI) [29], the National Primary Health Care Development Agency with support from the WHO to address the spread of cVDPV2 [25,30], the CORE Group Partners Project (CGPP) with the aid of the WHO, and the United Nations Children’s Fund (UNICEF) focusing on the health and safety concerns of northern communities and improving polio vaccine acceptance [17,31,32]. In February 2014, with WPV still being transmitted endemically in Pakistan, Afghanistan and Nigeria, the Nigerian health ministry introduced the inactivated polio vaccine (IPV) into the nation’s routine immunization schedule to children 14 weeks to 59 months of age [33].

However, vaccination services for polio in Nigeria fall short of the WHO’s target of 90% national coverage, as 2.3 million children in Nigeria have not received routine polio immunization, the second highest number in the world [34]. These shortcomings resonate with wider routine immunization rates reporting that about 3.1 million Nigerian children (14% of global data) are estimated to be zero-dose or missed-dose, while children who are not vaccinated are at risk for vaccine-preventable diseases such as polio [34,35].

Multiple reasons have been suggested for low or delayed polio vaccine uptake and challenges with polio eradication in Nigeria. A complex interplay of factors are involved, including misconceptions and misunderstanding about the polio vaccine, a boycott in Northern Nigeria and vaccine hesitancy, illiteracy, areas of conflict such as the Boko Haram insurgency, challenges in reaching underserved communities, failure of the cold-chain system and, more recently, the impact of the COVID-19 pandemic that disrupted global polio vaccination campaigns [36,37,38,39,40,41,42].

Overall, Nigeria’s health system is underdeveloped, with inadequate and outdated medical facilities and equipment and overall poor quality of care, especially in rural communities, which further increases health inequalities, limiting accessibility to services [43,44,45]. Other problems include corruption, poor drug availability and challenges of affordability, as most patients in Nigeria obtain their medications through out-of-pocket costs [29,46]. Hence, there are fundamental weaknesses in Nigeria’s health system and healthcare indicators remain some of the lowest in Africa [29,47,48,49].

Vaccine delivery in Nigeria is overseen by the NPI [29,50], with all vaccines approved by the WHO and delivered by trained health workers either at health facilities or at remote outreach sites in communities. Six different vaccines have been employed to stop polio transmission and for the eradication of WPV in Nigeria:▪Oral polio vaccine (OPV), i.e., trivalent oral polio vaccine (tOPV), which contains Sabin strains of all three poliovirus serotypes (i.e., OPV 1, 2 and 3), was used against poliovirus types 1, 2, and 3 and led to the eradication of WPV type 2 (WPV2) in Nigeria—first eradicated in 2015, with over 379 million OPV doses administered throughout Nigeria in 2013 and at least 200 million doses in 2014 [51,52].▪In April 2016, tOPV was withdrawn from routine vaccination coverage with the switch to bivalent oral polio vaccine (bOPV), which contains Sabin serotypes 1 and 3 strains (i.e., OPV1 and OPV3), protecting against poliovirus types 1 and 3 [51,53].▪To phase out type 2 OPV and provide protection against poliovirus types 1, 2 and 3, Nigeria incorporated a single dose of inactivated polio vaccine (IPV) at 14 weeks of age into the regular vaccination schedule in 2015.▪Monovalent oral polio vaccines types 1-2-3 (mOPV1, mOPV2 and mOPV3) were introduced to protect against each individual type of poliovirus, respectively. However, there were continuous outbreaks of cVDPV2 persisting in Nigeria despite the 180 million doses of mOPV2 administered between 2018 and 2022 [51,54]. Between 2018 and 2022, 526 cases of cVDPV2 were detected in Nigeria with spread to other countries within Africa [34,52].

Supplementary immunization activities (SIAs) delivered 16 million full doses and 19 million fractional doses of IPV between 2017 and 2020 to support the monovalent oral polio vaccine type 2 (mOPV2) outbreak response campaigns and increase IPV coverage [55]. By 2022, national coverage had grown from 42% at the time of commencement in 2015 to 62% in 2021, and a second IPV dosage was introduced at the age of six weeks [54,55].

▪In 2021, a novel oral poliovirus vaccine type 2 (nOPV2) was introduced, as Nigeria became the first country to be approved by the Global nOPV2 Advisory Group to use the more genetically stable nOPV2 vaccine, which has been administered to over 143.6 million children in Nigeria [52,54,56], enhancing routine immunization, with the aim of maintaining poliovirus immunity and preventing cVDPV2 outbreaks [54,55].

The recent outbreak and re-emergence of polio in Gaza is evidence that if circumstances disrupt routine immunization or public hygiene and sanitation in any country globally, there is a risk of polio thriving in that region. According to the WHO, children in all countries are at risk of polio as long as a single child remains infected, as the virus can easily be imported into a polio-free country [21,57]. For example, thousands of cVDPV cases in Nigeria have been responsible for the infection of people in 23 other countries, with population density contributing significantly to the spread in the three countries [47,58].

This review aimed to identify existing policies, strategies and interventions for polio eradication in Nigeria and their impact on polio vaccine uptake and eradication efforts in the country.

## 2. Materials and Methods

### 2.1. Search Strategy

An adapted systematic literature review was conducted in April 2024 using Ovid Medline and PubMed databases, which are the world’s leading bibliographic sources for biomedical and life sciences scholarly literature and research resources with the aim of improving health globally [59,60], alongside gray literature from Google Scholar not indexed in the Ovid Medline and PubMed databases, to identify articles reporting on the impact of polio vaccination services, compliance with immunization schedules, vaccine hesitancy and utilization of intervention policies directed towards polio eradication in Nigeria (see Figure 1).

Two databases were considered for the literature search to ensure comprehensive coverage of relevant articles [61,62]. Both published and gray literature were included to enhance the evidence base and minimize the impact of publication bias, which can occur when mainly research studies with statistically significant findings are published, as compared to less statistically significant results [63]. Potential limitations of the search strategy are documented in Box 1.
Box 1.Limitations of search strategyExclusion of literature (n = 261) focusing on subgroups or unrelated disease conditions and interventions. For example, the impact of COVID-19 on polio eradication services was excluded.Limiting the search to articles written in English. The present study excluded research published in other languages.Reliance on specific keywords in the PICO framework might miss articles that use different terminology or synonyms not included in the search terms. The search was focused on studies from 2018 to 2023; the study required comprehensive searches and adequate data from previous years, before 2018, ensuring that the study was based on recent and relevant research, which is in line with current evidence, practices and policies across the years.Some aspects of the PICO framework such as the “study design concept” were not relevant to the search strategy as the study is an exploratory, single-intervention study.
Figure 1Search strategy for literature to “identify existing policies, strategies, and interventions for polio eradication in Nigeria and their impact on polio vaccine uptake and eradication efforts in Nigeria”. Source: adapted from [64].
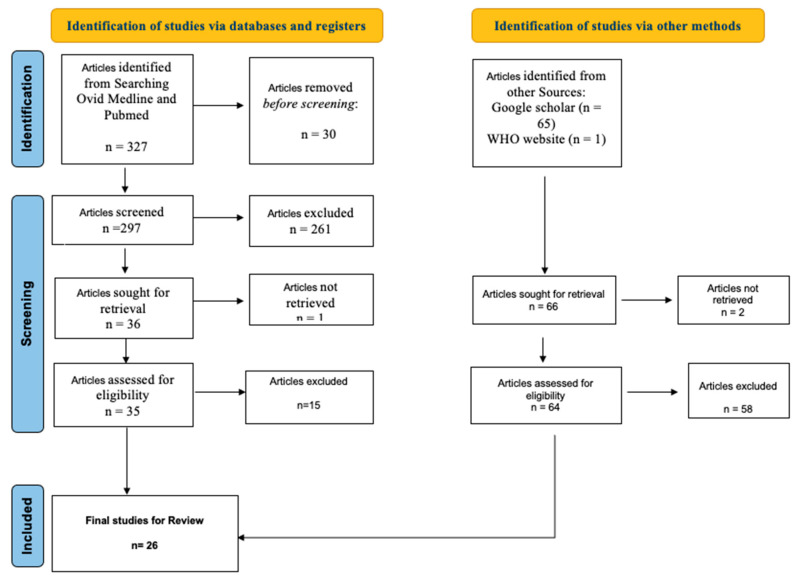


The review followed the key principles of systematic reviews, with literature identified using the most recent 2020 Preferred Reporting Items for Systematic Reviews and Meta-Analyses (PRISMA) Flowchart relating to database searches and the inclusion/exclusion process [64]. Search strategies used the Population, Intervention, Comparator and Outcome (PICO) framework [65] (Appendix A) to define key concepts, MeSH terms, MeSH tree subheadings and free search terms.

### 2.2. Selection Criteria

Inclusion and exclusion criteria were identified prior to conducting the search to reduce the risk of selection bias. Eligibility criteria included articles published in English after 2018, to ensure that the study was based on the most recent and relevant evidence, which is in line with current practices, policies and recommendations. The following search terms were used: “Impact”, “Polio”, “Nigeria”, “Polio vaccination services”, “Polio intervention policies” and “vaccine hesitancy’’, with the Ovid Medline and PubMed searches available in Appendix A. News or commentary articles were excluded. Full search Strategies for Pubmed literatures showing MeSH and free search terms generated for each PICO/S CONCEPT are available from the authors.

### 2.3. Data Extraction and Analysis/Synthesis

Data extraction and synthesis were conducted by OE using an Excel spreadsheet and discussed with SC. Themes were identified by thematic analysis of key issues across the 26 studies, using a synthesis table to represent central topics or subjects from the data sources.

The quality of the evidence in each study was assessed and appraised using the adapted Critical Appraisal Skills Program (CASP) [66]. The evidence summary table was modified from the Public Health Agency of Canada’s Control Guidelines Critical Appraisal Tool Kit [67].

## 3. Results

Six key themes were identified from the 26 studies included in the review: (1) polio intervention policies and programs; (2) polio vaccination services and mass campaigns; (3) vaccine uptake and vaccine hesitancy; (4) the Nigerian health system; (5) surveillance monitoring systems for polio eradication; and (6) challenges of polio vaccination services and interventions in terms of their impact on eradication efforts in Nigeria.

### 3.1. Polio Intervention Policies and Programs

Table 1 summarizes the polio intervention policies and programs identified in Nigeria. The Polio Eradication Initiative (PEI) enhanced health promotion, maternal and child health interventions, health worker skills, resource development, service delivery and system efficiency in health programs [31]. This included Emergency Operation Centers (EOCs) and improved data management systems, which supported the broader health system during outbreaks [31]. However, applying the PEI in a weak and understaffed health system led to role-shifting, reducing human resources efforts in rural areas and impacting service delivery [31]. Community engagement activities were important in Nigeria’s PEI as well as the need for well-structured implementation and continuous monitoring to ensure the effectiveness and sustainability of community engagement strategies [68].

The National Emergency Operation Centre (NEOC) and Data Working Group (DWG) were found to have played a crucial role in the Nigerian Polio Eradication Program [69,70], reducing the number of WPV1 cases from 122 in 2012 to zero by 2015, and enhancing its capabilities for managing other outbreaks like Ebola, measles, yellow fever and meningitis [69]. EOCs had enhanced coordination, strategic direction and disease control initiatives, but required significant financial support to maintain their functions [69].

The DWG provided quality surveillance and campaign information and assessed the preparedness for OPV SIAs. Independent monitoring and Lot Quality Assurance Sampling surveys identified low-performing areas for revaccination, with the data enabling the NEOC to make evidence-based decisions, identify gaps, improve preparedness and enhance routine immunization and surveillance activities [70].

The National Stop Transmission of Polio Program’s immunization program and eradication efforts contributed to improved immunization services in Nigeria, particularly in underserved areas [71], emphasizing the importance of targeted supervision and pre-campaign planning to enhance health campaign outcomes in regions with weak health campaigns [72].

The Core Group Partners Project (CGPP) and Volunteer Community Mobilizers (VCMs), introduced in Nigeria in 2014 to address the health and safety concerns of the northern communities, enhanced community engagement, helping to identify at risk groups and regions and improving polio vaccine acceptance [17,31,32]. Through the CGPP initiative, the number of WPV cases in Nigeria decreased from 1122 in 2006 to only 6 WPV cases in 2014 [30,73,74]. The training and empowerment of local volunteers highlighted the importance of community involvement and capacity building in public health systems, as VCMs improved vaccine acceptance by addressing community-specific concerns [73,75]. However, challenges included community resistance, misinformation, logistical challenges and security issues in high-risk areas [73].

The Nigerian government established the NPI with the primary goal of supporting the implementation of state and local government area immunization programs and providing free routine immunization [74,76]. In addition, the WHO, in collaboration with other partners such as the Bill & Melinda Gates Foundation and Gavi, the Vaccine Alliance, supported routine polio immunization, resulting in improved access to and coverage of routine immunization services [29]. The Nigerian NPI has, however, fallen short of achieving its goals due to inadequate infrastructure, lack of funding and poor service delivery, which have recently been exacerbated by the COVID-19 pandemic [50,76].

Nigeria’s polio eradication efforts were aided by international collaborations and partnerships. The WHO, through its transition framework, provided a comprehensive assessment of collaborative strategies emphasizing the importance of local and international partnerships with other countries and agencies, including, but not limited to, the WHO, UNICEF and the Bill & Melinda Gates Foundation [34]. Effective global collaboration transferred necessary resources and funding in support of polio eradication efforts and secured ongoing international commitment to assistance [39]. The WHO also provided vital insights into Nigeria’s polio transition plan to safeguard the essential polio functions, including poliovirus surveillance, immunization with appropriate polio vaccines, outbreak response and eradication efforts [34].
vaccines-13-00232-t001_Table 1Table 1Summary of polio intervention policies and programs in Nigeria.Intervention Policy/ProgramDescriptionImpactRelevant StudiesNational Immunization Days Regular nationwide vaccination campaignsIncreased coverage, reduced polio incidenceAdesola et al. [77],Braka et al. [69]Routine Immunization PolicyIncorporating polio vaccination into routine immunization schedulesImproved integration, sustained effortsBawa et al. [78], Erbeto et al. [70]Strategic Health PlansDevelopment of strategic plans at national and regional levelsSustained efforts, better resource allocationAkinyemi et al. [68], WHO [34]Mobile Health UnitsDeployment of mobile health units to reach underserved and remote populationsImproved access, increased coverageTaylor et al. [79], Akinyemi et al. [31]Community Health WorkersUtilization of community health workers for vaccine delivery and community mobilizationIncreased uptake, community educationAdamu et al. [17], Erbeto et al. [70]International Collaborations and PartnershipsCollaborations with international organizationsEnhanced support, sustained effortsWHO [34], Usman et al. [39]Supplementary Immunization Activities Targeted immunization campaigns in addition to routine immunizationsReduced incidence, improved herd immunityBiya et al. [71], Asekun et al. [56]Integration with Broader Health SystemsIntegrating polio vaccination with other health servicesBetter health outcomes, enhanced resilienceBawa et al. [78], Erbeto et al. [70]Culturally Sensitive Communication StrategiesTailoring communication to address cultural beliefs and misinformationReduced hesitancy, increased acceptanceOnoja [72], Edukugho et al. [72]Digital Health Tools and Surveillance SystemsUse of digital tools for better data collection and monitoringImproved detection, efficient managementCooper et al. [54], Shuab et al. [80]

### 3.2. Polio Vaccination Services: Frequency of Mass Campaigns and Immunization Coverage

Conflicting evidence was found on the impact of polio vaccination services providing frequent mass campaigns and immunization to boost polio eradication efforts in Nigeria. Extensive vaccination campaigns and high coverage of polio vaccination significantly reduced the incidence and eventually eliminated wild poliovirus in Nigeria. In 2020, Nigeria was declared free of wild poliovirus [75], with a 30% increase in polio vaccination rates, attributed to targeted vaccination services [31,79].

Community health workers played pivotal roles in increasing vaccine acceptance and coverage in local communities [17,70]. Other papers have emphasized the importance of community engagement and trust in the success of vaccination campaigns as data indicate a substantial increase in vaccination coverage in Northern and Southern Nigeria, especially in Northern Nigeria, where community engagement was previously low but rose from 45% in 2010 to 85% in 2020 [31,77].

However, integrating mass polio campaigns with routine health services in weaker health systems can divert resources away from other health programs, which frequently results in a lack of attention to basic health services and increased workload for healthcare workers [31,81]. Moreover, while mass vaccination campaigns and health systems through PEI implementation boosted health promotion activities and improved coverage for maternal and child health interventions [82], implementation within a weak health system causes a shift in health workers’ primary roles towards polio with potential neglect of other health programs due to resource diversion [31,81,82].

### 3.3. Vaccine Coverage and Vaccine Hesitancy

Attempts to eradicate polio in Nigeria have been significantly influenced by both coverage of vaccine schedules and vaccine hesitancy. Mass immunization campaigns increased the uptake of vaccines and including polio vaccination into routine immunization schedules significantly improved coverage, helping to sustain high immunization coverage over time [78]. Moreover, National Immunization Days effectively boosted immunization rates and compliance [77,78], and decreased polio cases due to higher vaccination compliance [39,70].

In January 2019, the WHO declared vaccine hesitancy as one of the top 10 leading threats to global public health [83]. In Nigeria, vaccine hesitancy due to cultural and religious beliefs and practices has been a significant factor influencing vaccination decisions. Cultural beliefs, misinformation and distrust in Western medicine were identified as reasons for vaccine hesitancy, especially in Northern Nigeria, which has hindered vaccination efforts [75,82,84].

Contrary to evidence about the success of mass immunization campaigns, there was also evidence of the unintended consequences of frequent campaigns, contributing to increased vaccine hesitancy in some communities. Some people felt overwhelmed by frequent campaigns, leading to a decrease in community participation, straining available resources in already weak health systems and resulting in lack of motivation among health workers [82].

### 3.4. The Nigerian Healthcare System

The co-delivery of polio vaccination services with other health interventions and services improved overall health outcomes and vaccination rates [51,70,77]. Integrating polio campaigns with other health services is important in helping to prevent the negative effects of frequent campaigns in a weak healthcare system and promoting community acceptance [79,82].

While initially a more focused approach to polio-specific activities, without integration with other health services, was advocated for [75], others have reported that the integration of other health services into polio eradication efforts ensured that health systems could respond more effectively to public health needs, improve vaccination uptake rates and enhance overall health service delivery [70,78].

Evidence showed that high-intensity immunization campaigns can cause fatigue, reduce motivation among health workers and also reduce community participation [31,51,82]. However, another study supports mass vaccination campaigns involving non-governmental organizations and volunteer community mobilizers as being cost-effective in high-risk areas [73]. Furthermore, modeling has shown that increasing the number of SIA rounds significantly reduced cVDPV cases, if effectively managed [55].

### 3.5. Surveillance Monitoring Systems in Polio Eradication Efforts

The National Emergency Operation Centre (NEOC) and Data Working Group (DWG) were responsible for the monitoring of and campaign information for the Nigerian Polio Eradication Program from 2016 to 2020 [70]. The NEOC used a pre-campaign dashboard to assess preparedness for supplementary immunization activities, for example, the use of acute flaccid paralysis surveillance data for case–control studies, with additional community controls surveyed from April 2021 [54,70]. Data indicated that >95% of children with non-polio acute flaccid paralysis had received >3 doses of OPV, highlighting high immunization coverage [54,70].

Active surveillance of adverse events of special interest was used to monitor the safety of nOPV2 in Nigeria [85]. Findings supported the continued monitoring of possible adverse events of special interest with the use of nOPV2 to combat polio. Active surveillance provided safety data, indicating a low incidence of adverse events [56,85].

### 3.6. Challenges to Polio Vaccination Services and Eradication Efforts in Nigeria

There have been various barriers to polio vaccination and eradication programs, emphasizing the importance of considering local contexts, community trust and effective communication in program implementation [86]. Barriers included a fear of vaccine side effects, distrust in the government and health services, poor infrastructure, negative cultural and religious beliefs and geopolitical instability [86].

Lack of resources and funding has impacted negatively on immunization coverage in Nigeria, as the general lack of basic necessities, including shortages of vaccines and supplies, instability of the cold chain, lack of training for health workers and volunteers, cancellation of immunization schedules and insecurity were identified as possible reasons for low coverage [86]. The WHO has reported a funding gap of US 132 million for effective polio eradication functions in Nigeria [34].

A report of the Center for Disease Control (CDC) reflected that from 2016 to 2017, there was an 87% reduction in the number of children in Nigeria without access to poliovirus vaccination compared to <60% in terms of national poliovirus vaccine coverage levels through routine immunization services since 2002 [17]. On the other hand, another study reported that the country experienced a setback in 2016 due to insecurity in northeastern Nigeria that started in 2012, restricting surveillance, immunization coverage and detection of WPV transmission, resulting in a resurgence of endemic WPV transmission in 2016 [87].

Over 60 polio campaigns were suspended by the GPEI during the COVID-19 pandemic, resulting in an increase in cases of cVDPV2 and outbreaks of cVDPV1 [47]. Evidently, COVID-19 caused gaps in polio detection and monitoring, and damaged health systems more broadly, including essential immunization programs.

## 4. Discussion and Future Directions

This review identified and assessed the impact of polio intervention services and policies on polio vaccine uptake and eradication efforts in Nigeria. Table 2 summarizes the key issues associated with low polio uptake that were identified in this paper, together with recommendations for changes to practice and policy to improve polio vaccination uptake in Nigeria and globally and recommendations for addressing these barriers in the short-term (1–2 years), mid-term (3–4 years) and long-term (5+ years). While the recommendations relate to improving polio uptake in Nigeria, they have transferable learning globally for all vaccination programs.

### 4.1. The Burden of Polio Globally and in Nigeria

The burden of WPV globally has fallen by 99%, from an estimated 350,000 cases in more than 125 endemic countries in 1988 to 12 cases in two endemic countries (Pakistan and Afghanistan) in August 2024, largely due to the efforts of the GPEI [10,58]. However, continued reports of cases of cVDPV in 2024 in Nigeria and the recent outbreak and re-emergence of paralytic polio in Gaza after 25 years show that if circumstances disrupt routine immunization, and there is a deterioration in hygiene and sanitation in any country, there is a risk of the highly infectious poliovirus thriving in that region, as it can easily be imported and spread within a polio-free country [21]. Moreover, while WPV has been successfully eradicated in Nigeria, there are ongoing risks for Nigeria with respect to both type 1 and type 3 vaccine-derived poliovirus outbreaks [47]. An outbreak of the type 1 virus in Nigeria could prolong interruption of the transmission of the VDPV by an extended period, and could pose a continuing threat to public health in Nigeria and other countries through international spread [97].

To reduce the burden of polio in Nigeria in the short-term, it is imperative to enhance immunization campaigns via SIAs, both generally and, in particular, in high-risk areas with low vaccination coverage and a high burden of polio, using the genetically stable nOPV2 to quickly control outbreaks of VDPV, as recommended by the Global nOPV2 Advisory Group [34,56]. There is a need to strengthen effective surveillance systems to improve early detection of poliovirus; for example, digital health tools can be used for real-time data collection and response, especially in hard-to-reach areas [56,85]. The reduction in polio cases from 2006 to 2014 shows the impact of surveillance systems in reaching unvaccinated populations and preventing resurgence of WPV [34,69,75,78]. The COVID-19 vaccine rollout demonstrated the need and opportunity to improve pharmacovigilance by integrating national vaccine platforms with surveillance across the African continent [89].

Community groups can encourage vaccine uptake by engaging community members to participate in local health campaigns and immunization programs [17,31]. The WHO vaccine advisory group recommended that community health workers and religious and community leaders can help address cultural concerns and misinformation about vaccines to encourage vaccine acceptance [83]. Neel et al. [82] observed similar reductions in polio cases in Ethiopia and India through high-coverage campaigns by community health workers.

In the medium term, there is a need to increase routine immunization coverage to reduce the polio burden in low vaccinated areas, necessitating an expansion of the vaccine supply to ensure full vaccination coverage and to ensure that routine immunization programs reach at least 90% of <5 year-old children [88]. This will require improvements to health system and workforce infrastructure in Nigeria to minimize the risk of infectious diseases and to reduce disease burden over time [98]. Increasing healthcare service funding and utilizing the public–private partnership (PPP) model, targeting non-profit, private and government partnerships that promote polio intervention services, could reduce the burden of polio in Nigeria [48]. There is also a need to monitor and contain cross-border transmission by cross-border surveillance and collaborative immunization campaigns between neighboring countries to prevent the spread of poliovirus across borders [47].

This review has found that sustained planning is required to embed improvements, with long-term recommendations including the need to maintain high immunization coverage and the development of policies that enhance the effectiveness of mass campaigns and promote the integration of campaign strategies with routine health services, thereby reducing the polio burden [81]. Such long-term planning requires the retention of polio eradication infrastructure and strategies to re-use various polio response frameworks to maintain preparedness for infectious disease outbreaks; for instance, the polio National Emergency Operation Centers (EOCs) could be transitioned to support broader public health functions [69]. Funding would need to be sustained both in Nigeria and globally, with global collaborative funding to combat infectious diseases as well as ongoing surveillance and disease response activities [75,99]. Evidence from Afghanistan and Pakistan shows that collaboration with neighboring countries has reduced WPV burden across borders [47]. Finally, there is a need for further and ongoing research to explore the impact of the burden of polio and other infectious diseases on health systems [82]; for example, to support the continued use of nOPV2 to reduce the burden of polio [85].

### 4.2. Existing Policies, Strategies and Interventions and Their Impact on Polio Eradication in Nigeria

Polio policies, frameworks and strategic plans (Table 1) have impacted polio eradication efforts and enhanced public health outcomes in Nigeria. For instance, the WHO Polio Transition Plan outlined strategies for integrating polio activities into the Nigerian health system with respect to the eradication of all forms of polio in Nigeria [34,69,77]. Similarly, the GPEI strategies were pivotal in reducing WPV transmission [31], while the introduction of the National Polio Emergency Action Plan, routine immunization policies and SIAs significantly enhanced the incorporation of polio vaccination services into routine immunization schedules and improved integration with routine health services [29,78,90].

Collaborative partnerships have supported the polio eradication efforts of the Nigerian government and, for example, the Bill & Melinda Gates Foundation, the WHO, the CDC, UNICEF, Gavi and Rotary International, boosting the confidence of the public with respect to vaccinations and mobilizing resources and funding for mass immunization campaigns, surveillance activities and coordination of eradication efforts [82]. Evidence from interventions in polio-endemic regions like Afghanistan and Pakistan shows that collaboration with GPEI partners and neighboring countries, augmented by community engagement and capacity building, has been important in preventing the spread of wild poliovirus across borders. But such efforts require regional collaboration from other countries in the provision of resources, funding and advocacy with respect to the eradication of polio in all affected countries [47,92,93].

Routine vaccination programs and National Immunization Days increased the uptake of vaccines and improved polio eradication efforts [77,78], with vaccine compliance reducing polio incidence and improving broader health outcomes by preventing other vaccine-preventable diseases through integrated health services [39]. Vaccine hesitancy, one of the WHO’s top ten threats to global public health [82], along with a polio vaccine boycott, undermined polio intervention policies and immunization coverage, especially in Northern Nigeria [84], which posed challenges in terms of maintaining population immunity against poliovirus [17]. Community-based intervention groups such as VCMs and the CGPP utilized community engagement strategies to help to improve vaccination uptake and address health safety acceptance, demonstrating the critical role of culturally sensitive approaches in public health interventions [17,74,75]. Community participation in mobilizing community members to participate in intervention programs is integral to promoting vaccine compliance and acceptance of intervention programs. [82,86]. However, vaccine hesitancy remains a significant barrier to the successful eradication of polio in Nigeria and is the major contributory factor to the re-emergence of cVDPV2 [84], underpinned by multiple complex contributors, including cultural, religious, security and political factors, particularly in Northern Nigeria [70,84,86]. Other factors, relating to the WHO SAGE Working Group 3C (convenience, complacency and confidence) model of vaccine hesitancy [100], included insecurity in hard-to-reach areas, lack of funding, government ownership, low motivation among health workers and poor resource management [71,72,78], as well as community resistance, misinformation, population displacement due to conflicts and variations in data collection [86]. Addressing misinformation and vaccine hesitancy is crucial for the success of vaccination programs, and ensuring high coverage of vaccinations to achieve herd immunity is essential [86]. Effective engagement of communities and education about the safety of vaccines are required to promote vaccine acceptance and to overcome cultural and informational challenges that contribute to vaccine hesitancy [17,96].

The delivery of polio intervention services in conjunction with other healthcare services within the broader health system can also improve overall health outcomes [51,70,77] and help health systems improve vaccination rates and other public health services [70,78]. For example, co-delivery of healthcare services such as distribution of treated mosquito nets and Vitamin A supplements to nursing mothers with other polio vaccination services promoted community participation and acceptance of intervention, as recommended by the WHO [34,82]. However, frequent mass vaccination campaigns can increase health workers’ workload and result in fatigue if not effectively integrated with broader health services [82,86]. An overwhelming focus on polio eradication has the potential of straining health system infrastructure and diverting resources away from other health services [31,75,81]; for example, limiting vaccination coverage and the health system’s capacity to respond to other health emergencies in Afghanistan and Pakistan [47].

The COVID-19 pandemic affected Nigeria’s polio eradication programs and highlighted limitations and weaknesses in the healthcare system [101]. For example, the COVID-19 vaccine rollout coincided with responses to the cVDPV2 outbreak. State-level resources were insufficient to oversee polio outbreak response campaigns and maintain cold-chain systems in the different states, with vaccination teams unable to reach some eligible children under 5 years of age in remote areas—a situation aggravated by ongoing security threats due to the insurgency in Nigeria [17,87]. In addition, the monopoly on distribution of nOPV2 by a single manufacturer delayed some vaccine distribution to Nigeria [56].

## 5. Conclusions

The aim of this review was to evaluate the impact of polio intervention services and policies on polio vaccine uptake and eradication efforts in Nigeria, which, after years of multi-sectoral polio eradication efforts, eradicated WPV in 2020. Nigeria has one of the highest burdens of VDPV from genetic mutation of the type 2 component of OPV, which has caused transmission of cVDPV2, particularly in communities with low immunity and poor immunization coverage [52]. The persistence of cVDPV2 is aggravated by vaccine hesitancy caused by cultural and religious influences, insecurity and poor health infrastructure. Despite the public health success of vaccination generally, building public confidence in vaccines and improving systems to monitor safety while maintaining data security and patient privacy are essential [89].

This review has identified progress made through mass immunization campaigns, integration of polio intervention services into the broader health system, targeted community engagement and international collaboration to facilitate polio eradication strategies. However, multiple reasons have been identified for low or delayed polio vaccine uptake and the resulting challenges with global polio eradication, which adversely impact the control and eradication of infectious and communicable diseases such as polio [35,71,102]. These challenges involve a complex interplay of factors, including misconceptions and misunderstandings about the polio vaccine; lack of access to healthcare services; a boycott in Northern Nigeria and vaccine hesitancy; illiteracy; areas in conflict, such as the area affected by the Boko Haram insurgency; challenges in reaching underserved communities; an ineffective vaccine service delivery system; deficits in surveillance and monitoring systems; low motivation among health workers; poor resource management and failures in the cold-chain system that have affected the delivery of vaccines in Nigeria; and, more recently, the impact of the COVID-19 pandemic, which disrupted global polio vaccination campaigns [36,37,38,39,40,41,42].

These experiences emphasize the need for healthcare system strengthening [98] and a sustained and funded focus on quality driven healthcare [101]. Also needed is a parallel focus on long-term workforce planning [94] and improved coordination at the national level among regulatory authorities and the department of health. Results from this review have important implications for public health practice and policy, both in Nigeria and globally. The recommendations and research priorities that have been suggested offer strategic advice for policymakers, researchers, public health professionals and clinical practitioners with respect to addressing the identified gaps and challenges. If implemented, the advice arising from this study would enhance the opportunities for Nigeria to sustain its WPD-free status and eradicate all variant forms of polio.

## Figures and Tables

**Table 2 vaccines-13-00232-t002:** Barriers and recommendations for practice and policy to improve polio vaccination uptake in Nigeria.

Barriers	Short-Term Recommendations	Medium-Term Recommendations	Long-Term Recommendations
Burden and monitoring of poliomyelitis	Maintain effective surveillance systems [56,85] Ongoing research on the burden of polio and geographic variations in Nigeria [82]	Expand global vaccine supply and immunization programs for <5 year-old children [88] and high-risk areas [79,86] Monitor and contain cross-border transmission by cross-border surveillance [47] Enhance surveillance systems in Nigeria [75] and across the African continent [89]	Embed community participation [39] Global vaccine development [54] Global public health emergency preparedness [56,77]
Polio intervention policies, services and workforce	Fund supplementary immunization activities [33,78,90] Maintain effective surveillance systems [56,85] Fund EOCs to maintain infectious disease outbreak preparedness [69] Maintain collaborative partnerships [22,77] Community participation [39] Ongoing research on interventions and implementation strategies with a data-driven approach [54,70,75,82]	Integrate and deliver polio services alongside other services [69,70,78,91] Effective workforce training [17,73,82] Maintain cross-border collaborations [92,93] Embed collaborative partnerships, i.e., Bill & Melinda Gates Foundation, UNICEF, WHO, CDC, Gavi [37,68,82] Expand CGPP and VCM community health worker programs [73,74] Encourage community participation [39,75]	5–10-year workforce strategy [73,82,94] Effective health system strengthening [85,91]
Vaccine uptake and vaccine hesitancy	Mass immunization campaigns [17] Culturally specific, health-literate community-based engagement [17,95,96] Research on vaccine safety and delivery and vaccine hesitancy [77] Research on polio vaccine-related adverse events [56] Train community members for vaccination campaigns [39,68,75,82,84]	Embed community members in vaccination campaigns [39,68,75,82,84] Non-governmental and volunteer community organizer collaboration (75)	Sustained immunization funding and mass vaccination campaigns [81] Continuous research on the socio-cultural dynamics influencing vaccine uptake [79] Research evaluating the long-term effectiveness and safety of polio vaccines [54]

## Data Availability

Additional data are available on reasonable request from the corresponding author. However, all informational sources and papers have been extensively referenced.

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
