# Peer review of "Poliomyelitis in Nigeria: Impact of Vaccination Services and Polio Intervention and Eradication Efforts"

_vaccines, 2025, doi:10.3390/vaccines13030232_

Round 1

Reviewer 1 Report

Comments and Suggestions for Authors

Thank you for the opportunity to review this manuscript ID: vaccines-3440521. This manuscript/review aimed to identify policies, strategies and interventions for polio eradication, and to assess their impact on polio vaccine uptake and eradication efforts in Nigeria.

For the most part, this manuscript presents the topic it deals with in a satisfactorily clear, informative and comprehensive manner. However, in some sections/subsections of this paper there are obvious omissions that require major revision.

In general, relevant literature is cited in the paper. Comment: References 111, 118 and 121 are cited in the paper, although the list of references at the end of the paper has 105 references; Check and correct this. Also, align the order of references in the text of the work.

The Introduction section presents the epidemiological situation and preventive strategies for poliomyelitis around the world, with a particularly detailed review of incidence patterns and previous achievements of preventive measures against poliomyelitis in Nigeria as one of the most populous populations in the world. The use of polio vaccines in Nigeria is described in detail. Current problems in the implementation of polio prevention and control measures in Nigeria are highlighted. At the end of the Introduction section, the objectives of this work are specified.

The Methods section describes in detail the procedures appropriate for this study design.

Comments:

- Not available:

`Supplementary Materials: The following supporting information can be downloaded at: www.mdpi.com/xxx/s1, Figure S1: title; Table S1: title; Video S1: title: Not applicable.`     

Correct this.

- Not available:

`Appendix A

Box A1: Literature Search Methodology Using the PICOS Framework

Table A1: Search Strategies For Ovid Medline Literature

Table A2: Search Strategies for Pubmed Literature.`      

Correct this.

In the Results section, the first key topic (`3.1: Polio intervention policies and programs`) is presented and described in detail.

Comments: Other key topics are not presented tabularly, only a description is given in the text. A tabular presentation of key topics 2-6 would significantly improve the results.

In the Discussion and future directions section, the burden of polio globally and in Nigeria is described in detail, with barriers and short-term, medium-term and long-term recommendations for practice and policy to improve polio vaccination uptake and to reduce the burden of polio in Nigeria.

Line 509: Subsection `Strengths and limitations of this study` is missing. Add this subsection with a discussion of the shortcomings of this study, as well as possibilities for their mitigation or elimination.

The Conclusions section highlighted the most important results of this study. Comments: Shorten the text of this section and avoid citing references in this section.     

Author Response

Author's Reply to the Review Report (Reviewer 1)

Please provide a point-by-point response to the reviewer’s comments and either enter it in the box below or upload it as a Word/PDF file. Please enter "Please see the attachment." in the box if you only upload an attachment. A template can be found here.

Comments and Suggestions for Authors

Thank you for the opportunity to review this manuscript ID: vaccines-3440521. This manuscript/review aimed to identify policies, strategies and interventions for polio eradication, and to assess their impact on polio vaccine uptake and eradication efforts in Nigeria.

For the most part, this manuscript presents the topic it deals with in a satisfactorily clear, informative and comprehensive manner. However, in some sections/subsections of this paper there are obvious omissions that require major revision.

Response: We thank the reviewer for their comment but disagree about “obvious omissions” and the reviewer does not suggest any references that are deemed to be absent.

In general, relevant literature is cited in the paper. Comment: References 111, 118 and 121 are cited in the paper, although the list of references at the end of the paper has 105 references; Check and correct this. Also, align the order of references in the text of the work.

Response: We have edited and reformatted the references, it was an oversight from previous editions of the work and we thank the reviewer. We have amended for references 111,118,121 and rearranged other citations to align in order of the references in the text.   

The Introduction section presents the epidemiological situation and preventive strategies for poliomyelitis around the world, with a particularly detailed review of incidence patterns and previous achievements of preventive measures against poliomyelitis in Nigeria as one of the most populous populations in the world. The use of polio vaccines in Nigeria is described in detail. Current problems in the implementation of polio prevention and control measures in Nigeria are highlighted. At the end of the Introduction section, the objectives of this work are specified.

Response: We thank the reviewer for this summary of the introduction.

The Methods section describes in detail the procedures appropriate for this study design.

Response: We thank the reviewer for this description of the methods.

Supplementary Materials: The following supporting information can be downloaded at: www.mdpi.com/xxx/s1, Figure S1: title; Table S1: title; Video S1: title: Not applicable.`     

Correct this.

Response: We have revised the Appendices as advised.

Appendix A

Box A1: Literature Search Methodology Using the PICOS Framework

Table A1: Search Strategies For Ovid Medline Literature

Table A2: Search Strategies for Pubmed Literature.`      

Correct this.

Response: We have made corrections and revised the Appendix materials as advised.

In the Results section, the first key topic (`3.1: Polio intervention policies and programs`) is presented and described in detail.

Comments: Other key topics are not presented tabularly, only a description is given in the text. A tabular presentation of key topics 2-6 would significantly improve the results.

Response: We thank the reviewer but believe that topic 3.1 has priority for tabulation in Table 1 as it summarizes the very interventions and policies that underpin the aims of the paper. We believe that tabulating all other topics would exceed the Journal Tables limit and make the paper less readable and more congested for the reader. The key factors from Topics 2-6 are tabulated in our short-medium-long term recommendations in Table 2.

In the Discussion and future directions section, the burden of polio globally and in Nigeria is described in detail, with barriers and short-term, medium-term and long-term recommendations for practice and policy to improve polio vaccination uptake and to reduce the burden of polio in Nigeria.

Response: We thank the reviewer for this summary of the discussion.

Line 509: Subsection `Strengths and limitations of this study` is missing. Add this subsection with a discussion of the shortcomings of this study, as well as possibilities for their mitigation or elimination.

Response: We followed the format stipulated by the journal. Box 1 highlights limitations of the search strategy.

The Conclusions section highlighted the most important results of this study. Comments: Shorten the text of this section and avoid citing references in this section.     

 Response: We thank the reviewer for this comment but believe the conclusions are of appropriate length and would ask the Assigned Editor to comment. We believe the citation of references reinforces key points and having the citations further strengthens the points to the readers.

Submission Date: 07 January 2025

Date of this review: 19 Jan 2025 18:42:24

Bottom of Form

© 1996-2025 MDPI (Basel, Switzerland) unless otherwise stated

Disclaimer Terms and Conditions Privacy Policy

Reviewer 2 Report

Comments and Suggestions for Authors

The review is comprehensive. I just have several minor suggestions.

1. Please revise the title to avoid repeatation.

2. Please add one figure to show the timeline of intervention and the epidemiology of polio with time.

Author Response

Author's Reply to the Review Report (Reviewer 2)

Please provide a point-by-point response to the reviewer’s comments and either enter it in the box below or upload it as a Word/PDF file. Please enter "Please see the attachment." in the box if you only upload an attachment. A template can be found here.

Comments and Suggestions for Authors

The review is comprehensive. I just have several minor suggestions.

  1. Please revise the title to avoid repetition.

Response: We revised the title to:

Poliomyelitis in Nigeria: Impact of Vaccination Services and Polio Intervention and Eradication Efforts.

  1. Please add one figure to show the timeline of intervention and the epidemiology of polio with time.

Response: We thank the reviewer for this comment but believe that the text in the introduction, especially on p.3 provides information on timelines of interventions.

Submission Date: 07 January 2025

Date of this review: 15 Jan 2025 12:22:32

Reviewer 3 Report

Comments and Suggestions for Authors

the paper is written very well

You provide short and medium term recommendations. What are the long term recommendations?

How cant he policy approaches you present in the paper be adapted for other diseases?

Author Response

Comments and Suggestions for Authors

The paper is written very well

Response:  We thank the author for this comment.

You provide short and medium term recommendations. What are the long term recommendations?

Response: Long-term recommendations are provided in Table 3 and discussion in the discussion.

How can the policy approaches you present in the paper be adapted for other diseases?

Response: We thank the author for this comment and as stated in the paper, we believe there is transferable learning for other diseases but that was not a main aim of the paper. We state on p.13 that “Finally, there is a need for further and ongoing research on the polio burden to explore the impact of polio and other infectious diseases burden on health systems”.

Submission Date: 07 January 2025

Date of this review: 15 Jan 2025 18:45:13

Round 2

Reviewer 1 Report

Comments and Suggestions for Authors

Thank you for the opportunity to re-review manuscript ID: vaccines-3440521.
The authors responded to my comments and made some corrections to this paper.
Note: as the authors themselves stated, regarding my comment on the Conclusions of this paper (article type: systematic review), the authors will ask the Assigned Editor.